# Randomized Entity-wise Factorization for Multi-Agent Reinforcement Learning

## Abstract

Real world multi-agent tasks often involve varying types and quantities of agents and non-agent entities; however, agents within these tasks rarely need to consider all others at all times in order to act effectively. Factored value function approaches have historically leveraged such independences to improve learning efficiency, but these approaches typically rely on domain knowledge to select fixed subsets of state features to include in each factor. We propose to utilize value function factoring with random subsets of entities in each factor as an auxiliary objective in order to disentangle value predictions from irrelevant entities. This factoring approach is instantiated through a simple attention mechanism masking procedure. We hypothesize that such an approach helps agents learn more effectively in multi-agent settings by discovering common trajectories across episodes within sub-groups of agents/entities. Our approach, **R**andomized **E**ntity-wise **F**actorization for **I**magined **L**earning (**REFIL**), outperforms all strong baselines by a significant margin in challenging StarCraft micromanagement tasks.

## 1 Introduction

Many real-world multi-agent tasks contain scenarios in which an agent must deal with varying numbers and/or types of cooperative agents, antagonist enemies or other entities. Agents, however, can often select their optimal actions while ignoring a subset of agents/entities. For example, in the sport of soccer, a "breakaway" occurs when an attacker with the ball passes the defense and only needs to beat the goalkeeper in order to score (see Figure 1). In this situation, only the opposing goalkeeper is immediately relevant to the attacker's success, so the attacker can safely ignore players other than the goalkeeper for the time being. By ignoring irrelevant context, the attacker can generalize this experience better to its next breakaway. Furthermore, soccer takes many forms, from casual 5 vs. 5 to full scale 11 vs. 11 matches, and breakaways occur in all. If agents can identify independent patterns of

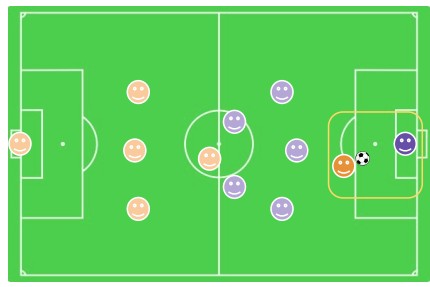

Figure 1: Breakaway sub-scenario in soccer. Agents in the yellow square can ignore the context outside of this region and still predict their success effectively.

behavior such as breakaways, they should be able to learn more efficiently as well as share their experiences across *all* forms of soccer.

Value function factoring approaches attempt to leverage independences between agents, such as those in our soccer example, by learning value functions as a combination of independent factors that depend on disjunct subsets of the state and action spaces (Koller & Parr, 1999). These subsets are typically fixed in advance using domain knowledge about the problem at hand, and thus are not scalable to complex domains where dependencies are unknown and may shift over time. Recent approaches in cooperative deep multi-agent reinforcement learning (MARL) factor value functions into separate components for each agent's action and observation space in order to enable decentralized execution (e.g., VDN (Sunehag et al., 2018), QMIX (Rashid et al., 2018)). These approaches learn a utility function for each agent that only depends on the agent's own action and its observations. The global $Q$-value is then predicted as some monotonic combination of these utilities in order to allow agents to greedily select their actions with local information while maximizing the

global $Q$. These approaches are able to effectively leverage independence between agents' local actions and observations, however, we note that observable entities are provided by the environment and are not all necessarily relevant to an agent's value function.

We build on these recent approaches by additionally factoring the observation space of each agent into factors for sub-groups of observed entities. Unlike classic works which factor the state or observation spaces, our work does not depend on fixed subsets of features designated through domain knowledge. Instead, we propose to *randomly* select sub-groups of observed entities and "imagine" the predicted utilities within these groups for each agent. These terms will not account for potential interactions outside of the groups, so we include additional factors that estimate the effect of the entities outside of each sub-group on each agent's utility. In order to estimate the true returns, we combine all factors using a mixing network (as in QMIX, Rashid et al., 2018), which allows our model to weight factors based on the *full state* context. We hypothesize this approach is beneficial for two reasons: 1) randomly partitioning entities and predicting returns from disjunct factors allows our model to explore *all* possible independence relationships among agents and entities, teaching agents to ignore irrelevant context when possible and 2) by teaching our models when they can ignore irrelevant context, they will learn more efficiently across varied settings that share common patterns of behavior, such as breakaways in soccer. The loss for training randomized factorization is added to the QMIX loss (i.e., using full observations) as an auxiliary objective. Our reasoning is again twofold: 1) we must learn the true returns to use as a target prediction for a Q-learning loss. 2) we do not know a priori which entities are unnecessary and thus need to learn policies that act on full observations.

Our entity-wise factoring procedure can be implemented easily in practice by using a simple masking procedure in attention-based models. Furthermore, by leveraging attention models, we can apply our approach to domains with varying entity quantities. Just as a soccer agent experiencing a breakaway can generalize their behavior across settings (5 vs. 5, 11 vs. 11, etc.) if they ignore irrelevant context, we hypothesize that our approach will improve performance across settings with variable agent and entity configurations. We propose **R**andomized **E**ntity-wise **F**actorization for **I**magined **L**earning (**REFIL**) and test on complex StarCraft Multi-Agent Challenge (SMAC) (Samvelyan et al., 2019) tasks with varying agent types and quantities, finding it attains improved performance over state-of-the-art methods.

## 2 BACKGROUND AND PRELIMINARIES

In this work, we consider the *decentralized partially observable Markov decision process* (Dec-POMDP) (Oliehoek et al., 2016), which describes fully cooperative multi-agent tasks. Specifically, we utilize the setting of Dec-POMDPs with entities (Schroeder de Witt et al., 2019).

**Dec-POMDPs with Entities** are described as tuples: $(\mathbf{S}, \mathbf{U}, \mathbf{O}, P, r, \mathcal{E}, \mathcal{A}, \Phi, \mu)$. $\mathcal{E}$ is the set of entities in the environment. Each entity $e$ has a state representation $s^e$, and the global state is the set $\mathbf{s} = \{s^e | e \in \mathcal{E}\} \in \mathbf{S}$. Some entities can be agents $a \in \mathcal{A} \subseteq \mathcal{E}$. Non-agent entities are parts of the environment that are not controlled by learning policies (e.g., landmarks, obstacles, agents with fixed behavior). The state features of each entity comprise of two parts: $s^e = [f^e, \phi^e]$ where $f^e$ represents the description of an entity's current state (e.g., position, orientation, velocity, etc.) while $\phi^e \in \Phi$ represents the entity's type (e.g., outfield player, goalkeeper, etc.), of which there are a discrete set. An entity's type affects the state dynamics as well as the reward function and, importantly, it remains fixed for the duration of the entity's existence. Not all entities may be visible to each agent, so we define a binary observability mask: $\mu(s^a, s^e) \in \{1, 0\}$, where agents can always observe themselves $\mu(s^a, s^a) = 1, \forall a \in \mathcal{A}$. Thus, an agent's observation is defined as $o^a = \{s^e | \mu(s^a, s^e) = 1, e \in \mathcal{E}\} \in \mathbf{O}$. Each agent $a$ can execute actions $u^a$, and the joint action of all agents is denoted as $\mathbf{u} = \{u^a | a \in \mathcal{A}\} \in \mathbf{U}$. $P$ is the state transition function which defines the probability $P(\mathbf{s}'|\mathbf{s}, \mathbf{u})$. $r(\mathbf{s}, \mathbf{u})$ is the reward function which maps the global state and joint actions to a single scalar reward.

We do not consider entities being added during an episode, but they may become inactive (e.g., a unit dying in StarCraft) in which case they no longer affect transitions and rewards. Since $\mathbf{s}$ and $\mathbf{u}$ are sets, their ordering does not matter, and our modeling construct should account for this (e.g., by modeling with permutation invariance/equivariance (Lee et al., 2019)). In many domains, the set of entity types present $\{\phi^e | e \in \mathcal{E}\}$ is fixed across episodes. We are particularly interested in cases where quantity and types of entities are varied between episodes, as identifying independence relationships between entities is crucial to generalizing experience effectively in these cases.

**Learning for Dec-POMDPs**   We aim to learn a set of policies that maximize expected discounted reward (returns) in some MDP. $Q$-learning is specifically concerned with learning an accurate action-value function $Q^{\text{tot}}$ (defined below), and using this function to select the actions that maximize expected returns. The optimal $Q$-function for the Dec-POMDP setting is defined as:

$$
\begin{aligned}
Q^{\text{tot}}(\mathbf{s}, \mathbf{u}) &:= \mathbb{E}\Big[ \sum_{t=0}^{\infty} \gamma^t \, r(\mathbf{s}_t, \mathbf{u}_t) \,\Big|\, {\substack{\mathbf{s}_0 = \mathbf{s}, \quad \mathbf{u}_0 = \mathbf{u}, \quad \mathbf{s}_{t+1} \sim P(\cdot | \mathbf{s}_t, \mathbf{u}_t) \\ \mathbf{u}_{t+1} = \arg\max Q^{\text{tot}}(\mathbf{s}_{t+1}, \cdot)}} \Big] \\
&= r(\mathbf{s}, \mathbf{u}) + \gamma \, \mathbb{E}\Big[ \max Q^{\text{tot}}(\mathbf{s}', \cdot) \,\big|\, s' \sim P(\cdot | \mathbf{s}, \mathbf{u}) \Big].
\end{aligned} \tag{1}
$$

Partial observability is typically handled by using the history of actions and observations as a proxy for state, typically processed by a recurrent neural network (RNN, Hausknecht & Stone, 2015): $Q_\theta^{\text{tot}}(\boldsymbol{\tau}_t, \mathbf{u}_t) \approx Q^{\text{tot}}(\mathbf{s}_t, \mathbf{u}_t)$, where the trajectory (i.e., action observation history) is $\tau_t^a := (o_0^a, u_0^a, \dots, o_t^a)$ and $\boldsymbol{\tau}_t := \{\tau_t^a\}_{a \in \mathcal{A}}$.

Work in deep reinforcement learning (Mnih et al., 2015) has popularized the use of neural networks as function approximators for learning $Q$-functions that are trained by minimizing the loss function:

$$
\mathcal{L}(\theta) := \mathbb{E}\Big[ \big( \underbrace{r_t + \gamma Q_{\bar\theta}^{\text{tot}}(\boldsymbol{\tau}_{t+1}, \arg\max Q_\theta^{\text{tot}}(\boldsymbol{\tau}_{t+1}, \cdot))}_{y_t^{\text{tot}}} - Q_\theta^{\text{tot}}(\boldsymbol{\tau}_t, \mathbf{u}_t) \big)^2 \,\Big|\, (\boldsymbol{\tau}_t, \mathbf{u}_t, r_t, \boldsymbol{\tau}_{t+1}) \sim \mathcal{D} \Big], \tag{2}
$$

where $\bar\theta$ are the parameters of a target network that is copied from $\theta$ periodically to improve stability (Mnih et al., 2015) and $\mathcal{D}$ is a replay buffer (Lin, 1992) that stores transitions collected by an exploratory policy (typically $\epsilon$-greedy). Double deep $Q$-learning (van Hasselt et al., 2016) mitigates overestimation of the learned values by using actions that maximize $Q_\theta^{\text{tot}}$ for the target network $Q_{\bar\theta}^{\text{tot}}$.

**Value Function Factorization**   Centralized training for decentralized execution (CTDE) has been a major focus in recent efforts in deep multi-agent RL (Lowe et al., 2017; Foerster et al., 2018; Sunehag et al., 2018; Rashid et al., 2018; Iqbal & Sha, 2019). Some work achieves CTDE by introducing methods for factoring $Q$-functions into monotonic combinations of per-agent utilities, with each depending only on a single agent's history of actions and observations $Q^a(\tau^a, u^a)$. This factorization allows agents to independently maximize their local utility functions in a decentralized manner with their selected actions combining to form the optimal joint action. This factored representation can only represent a limited subset of all possible value functions (Böhmer et al., 2020); however, these methods tend to perform better empirically than those that learn unfactored joint action value functions, most likely because they exploit independence properties among agents (Oliehoek et al., 2008). Sunehag et al. (2018) introduce value decomposition networks (VDN) which decompose the total $Q$-value as a sum of per-agent utilities: $Q^{\text{tot}}(\boldsymbol{\tau}, \mathbf{u}) := \sum_a Q^a(\tau^a, u^a)$. QMIX (Rashid et al., 2018) extends this approach to use a more expressive factorization. We describe QMIX and how we build our randomized factorization approach on top of it in Section 3.1.

**Attention Mechanisms for MARL**   Attention models have recently generated intense interest due to their ability to incorporate information across large contexts, including in the MARL literature (Jiang & Lu, 2018; Iqbal & Sha, 2019; Long et al., 2020). Importantly for our purposes, they are able to process variable sized sets of fixed length vectors (in our case entities). At the core of these models is a parameterized transformation known as multi-head attention (Vaswani et al., 2017). This transformation allows entities to selectively extract information from other entities based on their local context.

We define $\boldsymbol{X}$ as a matrix where each row corresponds to an entity (either its state representation or a transformed representation of it). The global state $\mathbf{s}$ can be represented in matrix form as $\boldsymbol{X}^{\mathcal{E}}$ where $\boldsymbol{X}_{e,*} = s^e$. Our models consist of entity-wise feedforward layers (denoted as eFF($\boldsymbol{X}$)) and multi-head attention layers (denoted as MHA $(\mathcal{A}, \boldsymbol{X}, \boldsymbol{M})$). Entity-wise feedforward layers apply an identical linear transformation to all input entities. Multi-head attention layers serve as a mechanism to integrate information across entities. These take in three arguments: the set of agents for which to compute an output vector $\mathcal{A}$, the matrix $\boldsymbol{X} \in \mathbb{R}^{|\mathcal{E}| \times d}$ where $d$ is the dimensionality of the input representations, and a mask $\boldsymbol{M} \in \mathbb{R}^{|\mathcal{A}| \times |\mathcal{E}|}$. The layer outputs a matrix $\boldsymbol{H} \in \mathbb{R}^{|\mathcal{A}| \times h}$ where $h$ is the hidden dimension of the layer. The row $\boldsymbol{H}_{a,*}$ corresponds to a weighted sum of linearly transformed representations from all entities selected by agent $a$. Importantly, if the entry of the mask $\boldsymbol{M}_{a,e} = 0$, then entity $e$'s representation cannot be included in $\boldsymbol{H}_{a,*}$. Masking serves two important purposes for us: 1) It enables decentralized execution by providing the mask $\boldsymbol{M}_{a,e}^\mu = \mu(s^a, s^e)$, such that agents can only see entities observable by them in the environment, and 2) It enable us to "imagine" the returns among sub-groups of entities. We integrate entity-wise feedforward layers and multi-

head attention into QMIX in order to adapt it to settings where the number of agents and entities is variable and build our approach from there. The exact process of computing attention layers, as well as the specifics of our attention-augmented version of QMIX are described in detail in the Appendix.

# 3 RANDOMIZED ENTITY-WISE FACTORIZATION FOR IMAGINED LEARNING

We now introduce our method, **R**andomized **E**ntity-wise **F**actorization for **I**magined **L**earning (**REFIL**). As discussed in Section 2, value function factorization approaches for cooperative deep MARL are motivated by their ability to exploit independence between agents while enabling decentralized execution with centralized training. We note that an agent's choice of optimal actions is often independent of a subset of its observed entities (cf. soccer breakaway example from Section 1), in addition to the choice of other agents' actions. Furthermore, we conjecture that agents robust to irrelevant entities should be more effective in dynamic environments with variable numbers of agents, as they are better able to identify shared patterns of behavior (e.g., breakaways exist in all forms of soccer). We do not know a priori which entities an agent can disregard, so we must consider *all possible sub-groups* of entities. As such, we propose to factor value functions by imagining returns in *random* sub-groups.

## 3.1 METHOD

QMIX (Rashid et al., 2018) relaxes the representational constraints of VDN (Sunehag et al., 2018), by allowing the joint value function $Q^{\text{tot}}$ to be a non-linear monotonic function with respect to the agent-specific utilities $Q^a$: $Q^{\text{tot}} = g\big(Q^1(\tau^1, u^1; \theta_Q), \dots, Q^{|\mathcal{A}|}(\tau^{|\mathcal{A}|}, u^{|\mathcal{A}|}; \theta_Q); \theta_g\big)$. The parameters of the mixing function $\theta_g$ are generated by a hyper-network (Ha et al., 2017) conditioning on the global state $\mathbf{s}$: $\theta_g = h(\mathbf{s}; \theta_h)$. Every state can therefore have a different mixing function, but the mixing's monotonicity maintains decentralizability, as agents can maximize $Q^{\text{tot}}$ without communication. All parameters $\theta = \{\theta_Q, \theta_h\}$ are trained with the DQN loss of Equation 2. We extend QMIX with attention layers both to encode variable sized sets of entities observed by each per-agent utility $Q^a$ and to mix the utilities of all agents $a \in \mathcal{A}$. Partial observability is implemented by a mask $M^\mu_{ae} = \mu(s^a, s^e), \forall a \in \mathcal{A}, \forall e \in \mathcal{E}$ that is provided to attention layers as described in section 2.

Building on QMIX, for each agent we generate a separate utility that only observes the state features of agents within its randomly selected sub-group: $Q^a_I(\tau^a_I, u^a)$, as well as a term that accounts for interactions *outside* of its group: $Q^a_O(\tau^a_O, u^a)$, then mixing these $2n$ (2 for each agent) utilities to form $Q^{\text{tot}}$. Importantly, since the mixing network is generated by the *full state context*, our model can weight factors contextually. For example, if agent $a$'s sampled sub-group contains all relevant information to compute its utility such that $Q^a_I \approx Q^a$, then the mixing network can weight $Q^a_I$ more heavily than $Q^a_O$. Otherwise, the network learns to balance $Q^a_I$ and $Q^a_O$ for each agent, using the full state as context, in order to estimate $Q^{\text{tot}}$. We train with these random factorizations in addition to the original QMIX objective. Treating factorization as auxiliary task, rather than as a representational constraint, allows our model to retain the expressivity of QMIX value functions (without sub-group partitions) while exploiting the potential independence between agents and other entities. We note that our auxiliary objective is only used in training, and execution in the environment does not use random factorization.

## 3.2 IMPLEMENTATION

The mechanism behind our entity-wise factorization relies on a simple attention masking procedure. In order to compute in-group utilities $Q^a_I(\tau^a_I, u^a)$ and out-group utilities $Q^a_O(\tau^a_O, u^a)$, we first randomly partition all entities in $\mathcal{E}$ into two disjunct groups (held fixed for an episode), indicated by a random binary[1] vector $\boldsymbol{m} \in \{0, 1\}^{|\mathcal{E}|}$. The entry $\boldsymbol{m}_e$ determines whether entity $e$ is in the first group, and we can take the negation $\neg \boldsymbol{m}_e$ to represent whether $e$ is in the second group. The subset of all agents is denoted as $\boldsymbol{m}_\mathcal{A} := [m_a]_{a \in \mathcal{A}}$. From these vectors, we can construct attention masks $\boldsymbol{M} \in \mathbb{R}^{|\mathcal{A}| \times |\mathcal{E}|}$. For example, using the mask $\boldsymbol{M}^1 = \boldsymbol{m}_\mathcal{A} \boldsymbol{m}^\top$, would prevent agents in the first group from "seeing" outside their group since $\boldsymbol{M}^1_{a,e} = 1$ *only* if agent $a$ and entity $e$ are in the same group. This can be added to a similarly produced mask $\boldsymbol{M}^2 = \neg \boldsymbol{m}_\mathcal{A} \neg \boldsymbol{m}^\top$ to create $\boldsymbol{M}_I$, a mask that only allows all agents to see the entities within their distinct groups. We construct masks for agents to see within ($\boldsymbol{M}_I$) and out of ($\boldsymbol{M}_O$) their groups, then combine with observability masks $\boldsymbol{M}^\mu$ as such:

$$\boldsymbol{M}^\mu_I := \boldsymbol{M}^\mu \wedge \boldsymbol{M}_I \,, \boldsymbol{M}^\mu_O := \boldsymbol{M}^\mu \wedge \boldsymbol{M}_O \,, \text{with } \boldsymbol{M}_I := \boldsymbol{m}_\mathcal{A} \boldsymbol{m}^\top \vee \neg \boldsymbol{m}_\mathcal{A} \neg \boldsymbol{m}^\top \,, \boldsymbol{M}_O := \neg \boldsymbol{M}_I. \quad (3)$$

---

[1] We first draw $p \in (0, 1)$ uniformly, followed by $|\mathcal{E}|$ independent draws from a Bernoulli($p$) distribution.

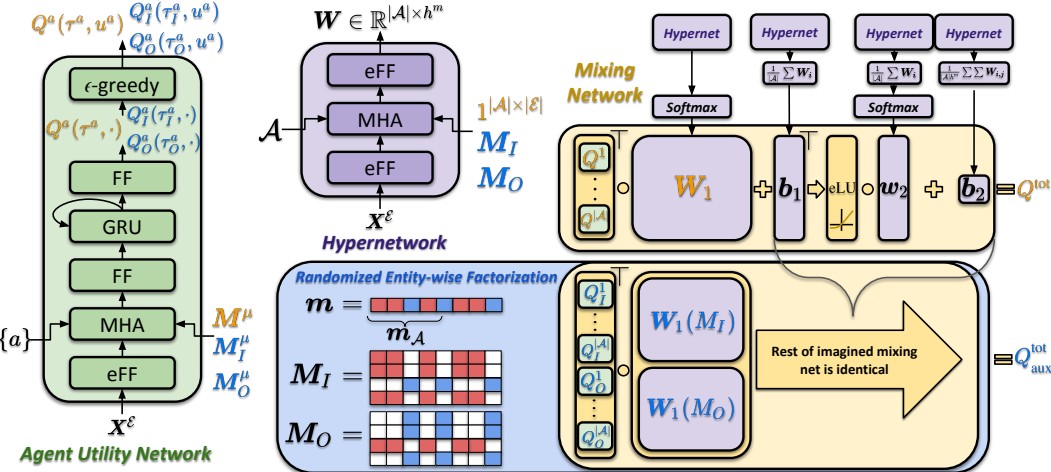

Figure 2: Schematic for **REFIL**. Values colored orange or blue are used for computing $Q^{\text{tot}}$ and $Q^{\text{tot}}_{\text{aux}}$ respectively. **(left)** Agent-specific utility networks. These are decentralizable due to the use of an observability mask ($\boldsymbol{M}^{\mu}$). We include Gated Recurrent Units (Chung et al., 2014) to retain information across timesteps in order to handle partial observability. **(top center)** Hypernetworks used to generate weights for the mixing network. We use a softmax function on the weights across the hidden dimension to enforce non-negativity, which we find empirically to be more stable than the standard absolute value function. Hypernetworks are not restricted by observability since they are only required during training and not execution. **(top right)** The mixing network used to calculate $Q^{\text{tot}}$. **(bottom right)** Procedure for performing randomized entity-wise factorization. For masks $M_I$ and $M_O$, colored spaces indicate a value of 1 (i.e., the agent designated by the row will be able to see the entity designated by the column), while white spaces indicate a value of 0. The color indicates which group the entity belongs to, so agents in the red group see red entities in $M_I$ and blue entities in $M_O$. Agents are split into sub-groups and their utilities are calculated for both interactions within their group, as well as to account for the interactions outside of their group, then monotonically mixed to predict $Q^{\text{tot}}_{\text{aux}}$.

The entry $\boldsymbol{M}^{\mu}_I[a, e]$ determines both whether agent $a$ can see entity $e$ and whether entity $e$ is in agent $a$'s group; the entry $\boldsymbol{M}^{\mu}_O[a, e]$ is the same but for entities out of $a$'s group. We can use these masks in our attention mechanisms to compute $Q^a_I(\tau^a_I, u^a)$, which represents the predicted utility of agent $a$ within its group and $Q^a_O(\tau^a_O, u^a)$, a residual term that accounts for the utility of interactions that $a$ would have with the other group.

Given each agent's predicted utility factors for both in-group and out-of-group, we combine these into a $Q^{\text{tot}}$ such that we can use the target from the full scenario ($y^{\text{tot}}_t$ in (2)) using a mixing network as in QMIX. This network's first layer typically takes $n$ inputs, one for each agent. Since we have $2n$ factors, we simply concatenate two generated versions of the input layer (using $\boldsymbol{M}_I$ and $\boldsymbol{M}_O$). We then apply the network to the concatenated utilities $Q^a_I(\tau^a_I, u^a)$ and $Q^a_O(\tau^a_O, u^a)$ of all agents $a$, to compute the predicted value $Q^{\text{tot}}_{\text{aux}}$. This procedure is visualized in Figure 2 and described in more detail in the Appendix.

Our novel approach **REFIL** uses $Q^{\text{tot}}_{\text{aux}}$ in place of $Q^{\text{tot}}$ in the DQN loss of (2) to get the auxiliary loss $\mathcal{L}_{\text{aux}}$. Our total loss combines both real and auxiliary losses: $\mathcal{L} := (1 - \lambda)\mathcal{L}_Q + \lambda \mathcal{L}_{\text{aux}}$, where $\lambda$ is a hyper-parameter. In practice, this procedure requires two additional passes through the network (with $\boldsymbol{M}^{\mu}_O$ and $\boldsymbol{M}^{\mu}_I$ as masks instead of $\boldsymbol{M}^{\mu}$) per training step. These additional passes can be parallelized by computing all necessary quantities in one batch on GPU. It is feasible to split entities into an arbitrary number $i$ of random sub-groups without using more computation by sampling several disjoint vectors $\boldsymbol{m}^i$ and combining them them in the same way as we combine $\boldsymbol{m}$ and $\neg \boldsymbol{m}$ in Equation 3 to form $\boldsymbol{M}_I$ and $\boldsymbol{M}_O$. Doing so could potentially bias agents towards considering smaller subsets of entities.

## 4 EXPERIMENTAL RESULTS

In our experiments, we aim to justify the main components of **REFIL**: 1) randomized sub-group factorization and 2) training as an auxiliary objective. We begin with experiments in a simple domain we construct such that agents' decisions rely only on a *subset* of all entities, and that subset is known, so we can compare our approach to approaches that use this domain knowledge. Then, we move

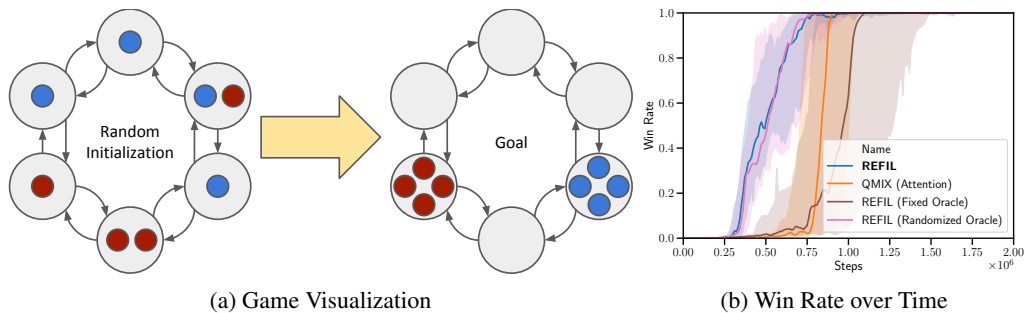

(a) Game Visualization              (b) Win Rate over Time

Figure 3: Group Matching Game. We use the values $n_a = 8$, $n_c = 6$, and $n_g = 2$ in our experiments. Shaded region is a 95% confidence interval across 24 runs.

on to testing on complex StarCraft micromanagement tasks to demonstrate our method's ability to scale to complex domains.

## 4.1 GROUP MATCHING GAME

We construct a group matching game, pictured in Figure 3a, where each agent only needs to consider a subset of other agents to act effectively and we know that subset as ground-truth (unlike in more complex domains such as StarCraft). As such, the task can be described as follows: Agents (of which there are $n_a$) are randomly placed in one of $n_c$ cells and assigned to one of $n_g$ groups (represented by the different colors) at the start of each episode. They can choose from three actions: move clockwise, stay, and move counter-clockwise. Their ultimate goal is to be located in the same cell as the rest of their group members, at which point an episode ends. There is no restriction on which cell agents form a group in (e.g., both groups can form in the same cell). All agents share a reward of 2.5 when any group is completed (and an equivalent penalty for a formed group breaking) as well as a penalty of -0.1 for each time step in order to encourage agents to solve the task as quickly as possible. Agents' entity-state descriptions $s^e$ include the cell that the agent is currently occupying as well as the group it belongs to (both one-hot encoded), and the task is fully-observable. Notably, agents can act optimally while ignoring agents outside of their group.

Ground-truth knowledge of relevant entities enables us to disentangle two aspects of our approach: the use of entity-wise factorization in general and specifically using randomly selected factors. We construct two approaches that use this knowledge to build factoring masks $M_I$ and $M_O$ which are used in place of randomly sampled groups (otherwise the methods are identical to **REFIL**). **REFIL** *(Fixed Oracle)* directly uses the ground truth group assignments (different at each episode) to build masks. **REFIL** *(Randomized Oracle)* randomly samples sub-groups from the ground truth groups only, rather than from all possible entities. We additionally train **REFIL** and *QMIX (Attention)* (i.e., **REFIL** with no auxiliary loss).

Figure 3b shows that using domain knowledge alone does not significantly improve performance in this domain (*QMIX (Attention)* vs. **REFIL** *(Fixed Oracle)*). In fact our *randomized* factorization approach is able to surpass the use of domain knowledge. The randomization in **REFIL** appears therefore to be crucial. One hypothesis for this phenomenon is that randomization of sub-group factors enables better knowledge sharing across diverse settings (in this case unique group assignments). For example, the situation of two agents from the same group being located in adjacent cells occurs within *all* possible group assignments. If sampling randomly, our approach will occasionally sample these two agents alone in their own group. Even if the rest of the context in a given episode has never been seen by the model before, as long as this sub-scenario has been seen, the model has some indication of the value associated with each action. Even when restricting the set of entities to form sub-groups with to those that we know can be relevant to each agent (**REFIL** *(Randomized Oracle)*) we find that performance does not significantly improve. These results suggest that randomized sub-group formation for **REFIL** is a viable strategy (vs attempting to learn which entities are relevant and selecting sub-groups from there), and the main benefit of our approach is to promote generalization across scenarios by breaking value function predictions into reusable components.

## 4.2 STARCRAFT

We next test on the StarCraft multi-agent challenge (SMAC) (Samvelyan et al., 2019). The tasks in SMAC involve micromanagement of units in order to defeat a set of enemy units in battle. Specifically, we extend SMAC to settings with variable types and quantities of agents. We hypothesize that our approach is especially beneficial in this setting, as it should encourage of models to identify independence between entities and generalize to more diverse settings as a result. The dynamic setting requires some small modifications to SMAC, though we aim to change the environment as little as possible to maintain the challenging nature of the tasks. In the standard version of SMAC, both state and action spaces depend on a fixed number of agents and enemies, so our modifications, discussed in detail in the appendix, alleviate these problems.

In our tests we evaluate on three settings we call 3-8sz, 3-8csz, and 3-8MMM. 3-8sz pits symmetrical teams of between 3 and 8 agents against each other where the agents are a combination of Zealots and Stalkers (similar to the 2s3z and 3s5z tasks in the original SMAC). 3-8csz pits symmetrical teams of between 0 and 2 Colossi and 3 to 6 Stalkers/Zealots against each other (similar to 1c3s5z). 3-8MMM pits symmetrical teams of between 0 and 2 Medics and 3 to 6 Marines/Marauders against each other (similar to MMM and MMM2). As a sanity check, we additionally modify our approach to work with non-attention models such that we can test on the original SMAC tasks against existing methods. These results (located in the appendix) show that we can significantly improve on QMIX (previously state-of-the-art) in 2 of 3 settings tested.

**Ablations and Baselines** We introduce several ablations of our method, as well as adaptations of existing methods to handle variable sized inputs. These comparisons are summarized in Table 1. *QMIX (Attention)* is our method without the auxiliary loss. **REFIL** (VDN) is our approach using summation to combine all factors (a la Value Decomposition Networks (Sunehag et al., 2018)) rather than a non-linear monotonic mixing network. *VDN (Attention)* does not include the auxiliary loss and uses summation as factor mixing. *QMIX (Mean Pooling)* is *QMIX (Attention)* with attention layers replaced by mean pooling. We also test max pooling but find the performance to be marginally worse than mean pooling. Importantly, for pooling layers we add entity-wise linear transformations prior to the pooling operations such that the total number of parameters is comparable to attention layers.

Table 1: Comparison of tested methods.

| Name | Imagined Learning | Entity Aggregation Method | Base Algorithm |
|---|---|---|---|
| **REFIL** | ✓ | MHA[1] | QMIX[2] |
| QMIX (Attention) | | MHA | QMIX |
| **REFIL** (VDN) | ✓ | MHA | VDN[3] |
| VDN (Attention) | | MHA | VDN |
| QMIX (Max Pooling) | | Max-Pool | QMIX |
| QMIX (EMP) | | EMP[4] | QMIX |
| ROMA (Attention) | | MHA | ROMA[5] |
| Qatten (Attention) | | MHA | Qatten[6] |
| QTRAN (Attention) | | MHA | QTRAN[7] |

[1]: Vaswani et al. (2017) [2]: Rashid et al. (2018) [3]: Sunehag et al. (2018)
[4]: Agarwal et al. (2019) [5]: Wang et al. (2020a) [6]: Yang et al. (2020)
[7]: Son et al. (2019)

For baselines we consider some follow-up works to QMIX that attempt to improve the mixing network to be more expressive: QTRAN (Son et al., 2019) and Qatten (Yang et al., 2020). We additionally consider an alternative mechanism for aggregating information across variable sets of entities, known as Entity Message Passing (EMP) (Agarwal et al., 2019). We specifically use the restricted communication setting where agents can only communicate with agents they observe, and we set the number of message passing steps to 3. Finally, we compare to a method that builds on QMIX by attempting to learn dynamic roles that depend on the context each agent observes: ROMA (Wang et al., 2020a). For all approaches designed for the standard SMAC setting, we extend them with the same multi-head attention architecture that our approach uses.

**Results and Discussion** Our results on challenges in dynamic STARCRAFT settings can be found in Figure 4. We find that **REFIL** outperforms all ablations consistently in these settings. **REFIL** *(VDN)* performs much worse than our approach and *VDN (Attention)*, highlighting the importance of the mixing network to handle contextual dependencies between entity partitions. Since the trajectory of a subset of entities can play out differently based on the surrounding context, it's important for our factorization approach to recognize and adjust for these situations. The mixing network handles these dependencies by a) incorporating global state information into the mixing procedure, and b) mixing utilities in a non-linear monotonic fashion, rather than summing as in VDN. As such, the increased representative capacity of the QMIX mixing network, relative to VDN, is crucial. The use of mean-pooling in place of attention also performs poorly, indicating that attention is valuable for aggregating information from variable length sets of entities.

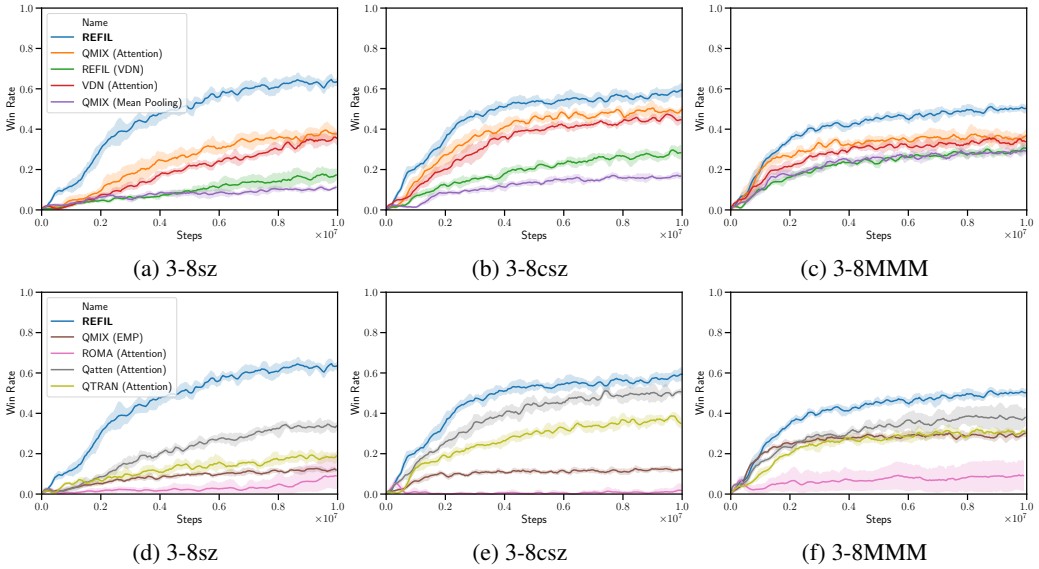

Figure 4: Test win rate over time on variable entity STARCRAFT environments. Shaded region is a 95% confidence interval across 6 runs. (top row) Ablations of our method. (bottom row) Baseline methods.

With respect to the baselines, we also find that **REFIL** consistently outperforms other methods, highlighting the unique challenge of learning in such dynamic settings where entity types are variable at each episode. The improvements that ROMA, Qatten, and QTRAN seen in other settings over QMIX, do not appear to manifest themselves in this setting. Moreover, the entity aggregation method of EMP does not improve performance over the standard MHA module that we use, likely due to the fact that EMP is most effective in settings where partial observability is a major hindrance to successful task completion. In this way, the target of EMP and **REFIL** are opposite, as the goal of **REFIL** is to ignore extraneous information when possible during training to improve knowledge transfer.

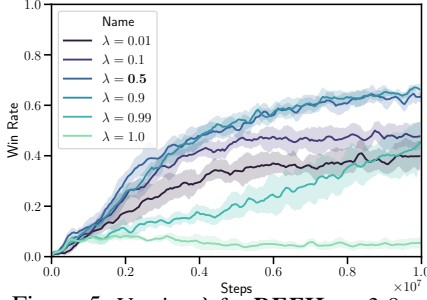

Figure 5: Varying $\lambda$ for **REFIL** on 3-8sz.

In order to understand the role of training as an auxiliary objective (rather than entirely replacing the objective) we vary the value of $\lambda$ to interpolate between two modes: $\lambda = 0$ is simply *QMIX (Attention)*, while $\lambda = 1$ trains exclusively with random factorization. Our results (Figure 5) show that, similar to regularization methods such as Dropout (Srivastava et al., 2014), there is a sweet spot where performance is maximized before collapsing catastrophically. Training exclusively with random factorization does not learn anything significant. This failure is likely due to the fact that we use the full context in our targets for learning with imagined scenarios as well as when executing our policies, so we still need to learn with it in training.

Finally, we consider a qualitative experiment to highlight the sort of common patterns that **REFIL** is able to leverage (Figure 6). Zealots (the only melee unit present) are weak to Colossi, so they learn to hang back and let other units engage first. Then, they jump in and intercept the enemy Zealots while all other enemy units are preoccupied, leading to a common pattern of a Zealot vs. Zealot skirmish (highlighted at t=15). **REFIL** enables behaviors learned in these types of sub-groups to be applied more effectively across all unique unit type configurations. By sampling groups from all entities randomly, we will occasionally end up with sub-groups that include only Zealots, and the value function predictions learned in these sub-groups can be applied not only to the episode at hand, but to any episode where a similar pattern emerges.

## 5  RELATED WORK

Multi-agent reinforcement learning (MARL) is a broad field encompassing cooperative (Foerster et al., 2018; Rashid et al., 2018; Sunehag et al., 2018), competitive (Bansal et al., 2018; Lanctot

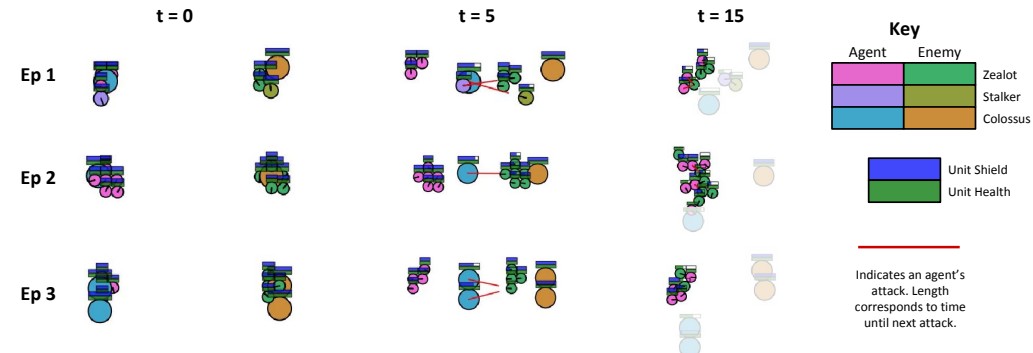

Figure 6: Simplified rendering of a common pattern that emerges across initializations in the 3-8csz SMAC setting, highlighted at $t = 15$. **REFIL** enables learning from each episode to inform behavior in the others.

et al., 2017), and mixed (Lowe et al., 2017; Iqbal & Sha, 2019) settings. This paper focuses on cooperative MARL with centralized training and decentralized execution (Oliehoek et al., 2016, CTDE). Our approach utilizes value function factorization, an approach aiming to simultaneously overcome limitations of both joint Hausknecht (2016) and independent learning Claus & Boutilier (1998) paradigms. Early attempts at value function factorisation require apriori knowledge of suitable per-agent team reward decompositions or interaction dependencies. These include optimising over local compositions of individual Q-value functions learnt from individual reward functions (Schneider et al., 1999), as well as summing individual $Q$-functions with individual rewards before greedy joint action selection (Russell & Zimdars, 2003). Guestrin et al. (2002); Kok & Vlassis (2006) factorise the total $Q$-value function using *coordination graphs* based on interaction dependencies for the task at hand, similarly to max-plus approaches Kuyer et al. (2008); Pol & Oliehoek (2016). Recent approaches from cooperative deep multi-agent RL allow for value factorisations to be learnt from experience from a single team reward function and no prior knowledge of interaction dependencies. Value-Decomposition Networks (VDN) (Sunehag et al., 2018) decompose the joint $Q$-value function into a sum of local utility functions used for greedy action selection. QMIX Rashid et al. (2018) extends such additive decompositions to general monotonic functions. Several works extend QMIX to improve the expressivity of mixing functions (Son et al., 2019; Yang et al., 2020), learn latent embeddings to help exploration (Mahajan et al., 2019) or learn dynamic roles (Wang et al., 2020a), and encode knowledge of action semantics into network architectures (Wang et al., 2020b).

Several recent works have addressed the topic of generalization and transfer across environments with varying agent quantities, though the learning paradigms considered and assumptions made differ from our approach. Carion et al. (2019) devise an approach for assigning agents to tasks, assuming the existence of low-level controllers to carry out the tasks, and show that it can scale to much larger scenarios than those seen in training. Burden (2020) propose a transfer learning approach using convolutional neural networks and grid-based state representations to scale to scenarios of arbitrary size. Agarwal et al. (2019) introduce an entity message passing framework to enable agents to attend to specific entities, of which there may be a variable amount, based on their local context, similar to the multi-head attention module we use in our approach. Several approaches devise attention or graph-neural-network based models for handling variable sized inputs and focus on learning curricula to progress on increasingly large/challenging settings (Long et al., 2020; Baker et al., 2019; Wang et al., 2020c). In contrast to these curriculum learning approaches, we focus on training simultaneously on scenarios of varying sizes and specifically focus on developing a training paradigm for improving knowledge sharing across such settings to accelerate learning.

## 6 CONCLUSION

In this paper we consider a MARL setting where we aim to learn policies to control teams of agents in scenarios with varying types and quantities of entities. We propose **REFIL**, an approach that regularizes value functions to identify independence relationships between entities, in turn promoting generalization and knowledge transfer within and across multi-agent settings with varying quantities of agents. Our results show that our contributions yield performance improvements in complex cooperative tasks. In future work, we hope to explore alternative methods for learning independence relationships between entities beyond randomized partitions.

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

## A    ATTENTION LAYERS AND MODELS

Attention models have recently generated intense interest due to their ability to incorporate information across large contexts. Importantly for our purposes, they are able to process variable sized sets of inputs.

We now formally define the building blocks of our attention models. Given the input $\boldsymbol{X}$, a matrix where the rows correspond to entities, we define an entity-wise feedforward layer as a standard fully connected layer that operates independently and identically over entities:

$$\text{eFF}(\boldsymbol{X}; \boldsymbol{W}, \boldsymbol{b}) = \boldsymbol{X}\boldsymbol{W} + \boldsymbol{b}^\top, \boldsymbol{X} \in \mathbb{R}^{n^x \times d}, \boldsymbol{W} \in \mathbb{R}^{d \times h}, \boldsymbol{b} \in \mathbb{R}^h \tag{4}$$

Now, we specify the operation that defines an attention head, given the additional inputs of $\mathcal{S} \subseteq \mathbb{Z}^{[1,n^x]}$, a set of indices that selects which rows of the input $\boldsymbol{X}$ are used to compute queries such that $X_\mathcal{S} \in \mathbb{R}^{|\mathcal{S}| \times d}$, and $\boldsymbol{M}$, a binary obserability mask specifying which entities each query entity can observe (i.e. $\boldsymbol{M}_{i,j} = 1$ when $i \in \mathcal{S}$ can incorporate information from $j \in \mathbb{Z}^{[1,n^x]}$ into its local context):

$$\text{Atten}(\mathcal{S}, \boldsymbol{X}, \boldsymbol{M}; \boldsymbol{W}^Q, \boldsymbol{W}^K, \boldsymbol{W}^V) = \text{softmax}\left(\text{mask}\left(\frac{\boldsymbol{Q}\boldsymbol{K}^\top}{\sqrt{h}}, \boldsymbol{M}\right)\right)\boldsymbol{V} \in \mathbb{R}^{|\mathcal{S}| \times h} \tag{5}$$

$$\boldsymbol{Q} = \boldsymbol{X}_{\mathcal{S},*}\boldsymbol{W}^Q, \boldsymbol{K} = \boldsymbol{X}\boldsymbol{W}^K, \boldsymbol{V} = \boldsymbol{X}\boldsymbol{W}^V, \quad \boldsymbol{M} \in \{0,1\}^{|\mathcal{S}| \times n^x}, \boldsymbol{W}^Q, \boldsymbol{W}^K, \boldsymbol{W}^V \in \mathbb{R}^{d \times h} \tag{6}$$

The mask$(\boldsymbol{Y}, \boldsymbol{M})$ operation takes two equal sized matrices and fills the entries of $\boldsymbol{Y}$ with $-\infty$ in the indices where $\boldsymbol{M}$ is equal to 0. After the softmax, these entries become zero, thus preventing the attention mechanism from attending to specific entities. This masking procedure is used in our case to uphold partial observability. Only one attention layer is permitted in the decentralized execution setting; otherwise information from unseen agents can be propagated through agents that are seen. $\boldsymbol{W}^Q$, $\boldsymbol{W}^K$, and $\boldsymbol{W}^V$ are all learnable parameters of this layer. Queries, $\boldsymbol{Q}$, can be thought of as vectors specifying the type of information that an entity would like to select from others, while keys, $\boldsymbol{K}$, can be thought of as specifying the type of information that an entity possesses, and finally, values, $\boldsymbol{V}$, hold the information that is actually shared with other entities.

We define multi-head-attention as the parallel computation of attention heads as such:

$$\text{MHA}(\mathcal{S}, \boldsymbol{X}, \boldsymbol{M}) = \text{concat}\left(\text{Atten}\left(\mathcal{S}, \boldsymbol{X}, \boldsymbol{M}; \boldsymbol{W}_j^Q, \boldsymbol{W}_j^K, \boldsymbol{W}_j^V\right), j \in \left(1 \ldots n^h\right)\right) \tag{7}$$

The size of the parameters of an attention layer does not depend on the number of input entities. Furthermore, we receive an output vector for each query vector.

## B    AUGMENTING QMIX WITH ATTENTION

The standard QMIX algorithm relies on a fixed number of entities in three places: inputs of the agent-specific utility functions $Q_a$, inputs of the hypernetwork, and the number of utilities entering the mixing network, that is, the output of the hypernetwork. QMIX uses multi-layer perceptrons for which all these quantities have to be of fixed size. In order to adapt QMIX to the variable agent quantity setting, such that we can apply a single model across all episodes, we require components that accept variable sized sets of entities as inputs. By utilizing attention mechanisms, we can design components that are no longer dependent on a fixed number of entities taken as input. We define the following inputs: $\boldsymbol{X}_{ei}^\mathcal{E} := s_i^e, 1 \le i \le d, e \in \mathcal{E}; \boldsymbol{M}_{ae}^\mu := \mu(s^a, s^e), a \in \mathcal{A}, e \in \mathcal{E}$. The matrix $\boldsymbol{X}^\mathcal{E}$ is the global state $\mathbf{s}$ reshaped into a matrix with a row for each entity, and $\boldsymbol{M}^\mu$ is a binary observability matrix which enables decentralized execution, determining which entities are visible to each agent.

### B.1    UTILITY NETWORKS

While the standard agent utility functions map a flat observation, whose size depends on the number of entities in the environment, to a utility for each action, our attention-utility functions can take in a variable sized set of entities and return a utility for each action. The attention layer output for agent $a$ is computed as MHA $(\{a\}, \boldsymbol{X}, \boldsymbol{M}^\mu)$, where $\boldsymbol{X}$ is an row-wise transformation of $\boldsymbol{X}^\mathcal{E}$ (e.g.,

an entity-wise feedforward layer). If agents share parameters, the layer can be computed in parallel for all agents by providing $\mathcal{A}$ instead of $\{a\}$, which we do in practice.

## B.2 Generating Dynamic Sized Mixing Networks

Another challenge in devising a QMIX algorithm for variable agent quantities is to adapt the hypernetworks that generate weights for the mixing network. Since the mixing network takes in utilities from each agent, we must generate feedforward mixing network parameters that change in size depending on the number of agents present, while incorporating global state information. Conveniently, the number of output vectors of a MHA layer depends on the cardinality of input set $\mathcal{S}$ and we can therefore generate mixing parameters of the correct size by using $\mathcal{S} = \mathcal{A}$ and concatenating the vectors to form a matrix with one dimension size depending on the number of agents and the other depending on the number of hidden dimensions. Attention-based QMIX (QMIX (Attention)) trains these models using the standard DQN loss in Equation 2.

Our two layer mixing network requires the following parameters to be generated: $\boldsymbol{W}_1 \in \mathbb{R}^{+(|\mathcal{A}| \times h^m)}$, $\boldsymbol{b}_1 \in \mathbb{R}^{h^m}$, $\boldsymbol{w}_2 \in \mathbb{R}^{+(h^m)}$, $b_2 \in \mathbb{R}$, where $h^m$ is the hidden dimension of the mixing network and $|\mathcal{A}|$ is the set of agents.

Note from Eq. (5) that the output size of the layer is dependent on the size of the query set. As such, using attention layers, we can generate a matrix of size $|\mathcal{A}| \times h^m$, by specifying the set of agents, $\mathcal{A}$, as the set of queries $\mathcal{S}$ from Eq. (5). We do not need observability masking since hypernetworks are only used during training and can be fully centralized. For each of the four components of the mixing network ($\boldsymbol{W}_1, \boldsymbol{b}_1, \boldsymbol{w}_2, b_2$), we introduce a hypernetwork that generates parameters of the correct size. Thus, for the parameters that are vectors ($\boldsymbol{b}_1$ and $\boldsymbol{w}_2$), we average the matrix generated by the attention layer across the $|\mathcal{A}|$ sized dimension, and for $b_2$, we average all elements. This procedure enables the dynamic generation of mixing networks whose input size varies with the number of agents. Assuming $\boldsymbol{q} = [Q^1(\tau^1, u^1), \ldots, Q^n(\tau^n, u^n)]$, then $Q^{\text{tot}}$ is computed as:

$$Q^{\text{tot}}(\mathbf{s}, \tau, \mathbf{u}) = \sigma((\boldsymbol{q}^\top \boldsymbol{W}_1) + \boldsymbol{b}_1^\top) \boldsymbol{w}_2 + b_2 \tag{8}$$

where $\sigma$ is an ELU nonlinearity (Clevert et al., 2015).

## C Environment Details

### C.1 StarCraft with Variable Agents and Enemies

The standard version of SMAC loads map files with pre-defined and fixed unit types, where the global state and observations are flat vectors with segments corresponding to each agent and enemy. Partial observability is implemented by zeroing out segments of the observations corresponding to unobserved agents. The size of these vectors changes depending on the number of agents placed in the map file. Furthermore, the action space consists of movement actions as well as separate actions to attack each enemy unit. As such the action space also changes as the number of agents changes.

Our version loads empty map files and programmatically generates agents, allowing greater flexibility in terms of the units present to begin each episode. The global state is split into a list of equal-sized entity descriptor vectors (for both agents and enemies), and partial observability is handled by generating a matrix that shows what entities are visible to each agent. The variable-sized action space is handled by randomly assigning each enemy a tag at the beginning of each episode and designating an action to attack each possible tag, of which there are a maximum number (i.e. the maximum possible number of enemies across all initializations). Agents are able to see the tag of the enemies they observe and can select the appropriate action that matches this tag in order to attack a specific enemy.

## D Experimental Details

Our experiments were performed on a desktop machine with a 6-core Intel Core i7-6800K CPU and 3 NVIDIA Titan Xp GPUs, and a server with 2 16-core Intel Xeon Gold 6154 CPUs and 10 NVIDIA Titan Xp GPUs. Each experiment is run with 8 parallel environments for data collection and a single GPU. **REFIL** takes about 24 hours to run for 10M steps on STARCRAFT. QMIX (Attention) takes

about 16 hours for the same number of steps on STARCRAFT. Reported times are on the desktop machine and the server runs approximately 15% faster due to more cores being available for running the environments in parallel.

## E    HYPERPARAMETERS

Hyperparameters were based on the PyMARL (Samvelyan et al., 2019) implementation of QMIX and are listed in Table 2. All hyperparameters are the same in all STARCRAFT settings. Since we train for 10 million timesteps (as opposed to the typical 2 million in standard SMAC), we extend the epsilon annealing period (for epsilon-greedy exploration) from 50,000 steps to 500,000 steps. For hyperparameters new to our approach (hidden dimensions of attention layers, number of attention heads, $\lambda$ weighting of imagined loss), the specified values in Table 2 were the first values tried, and we found them to work well. The robustness of our approach to hyperparameter settings, as well as the fact that we do not tune hyperparameters per environment, is a strong indicator of the general applicability of our method.

Table 2: Hyperparameter settings across all runs and algorithms/baselines.

| Name | Description | Value |
|------|-------------|-------|
| lr | learning rate | 0.0005 |
| optimizer | type of optimizer | RMSProp[1] |
| optim $\alpha$ | RMSProp param | 0.99 |
| optim $\epsilon$ | RMSProp param | $1e-5$ |
| target update interval | copy live params to target params every _ episodes | 200 |
| bs | batch size (# of episodes per batch) | 32 |
| grad clip | reduce global norm of gradients beyond this value | 10 |
| $|D|$ | maximum size of replay buffer (in episodes) | 5000 |
| $\gamma$ | discount factor | 0.99 |
| starting $\epsilon$ | starting value for exploraton rate annealing | 1.0 |
| ending $\epsilon$ | ending value for exploraton rate annealing | 0.05 |
| anneal time | number of steps to anneal exploration rate over | 500000 |
| $h^a$ | hidden dimensions for attention layers | 128 |
| $h^r$ | hidden dimensions for RNN layers | 64 |
| $h^m$ | hidden dimensions for mixing network | 32 |
| # attention heads | Number of attention heads | 4 |
| nonlinearity | type of nonlinearity (outside of mixing net) | ReLU |
| $\lambda$ | Weighting between standard QMIX loss and imagined loss | 0.5 |

[1]: Tieleman & Hinton (2012)

## F    ADDITIONAL RESULTS

We test a modified non-attention version of our approach along with state of the art methods on the standard version of SMAC, where entity types are constant at the start of each episode. Since the number and type of agents and enemies is constant at each episode, observations and states can be represented as fixed-size vectors. We can thus use MLPs as models (as is standard in the literature) for these tasks and adapt our approach to suit this setting while comparing to unmodified versions of existing approaches. Rather than masking an attention mechanism, we simply zero out the features in the observations and states that correspond to entities we would like to mask out. These experiments are performed in order to compare our approach to results validated in the literature.

We compare against QMIX (Rashid et al., 2018) and VDN (Sunehag et al., 2018), as well as an ablation of our approach that uses additive mixing (a la VDN) of entity partition factors instead of a mixing network which we call **REFIL** (VDN). We use the architectures and hyperparameters from the QMIX (Rashid et al., 2018) paper in these settings.

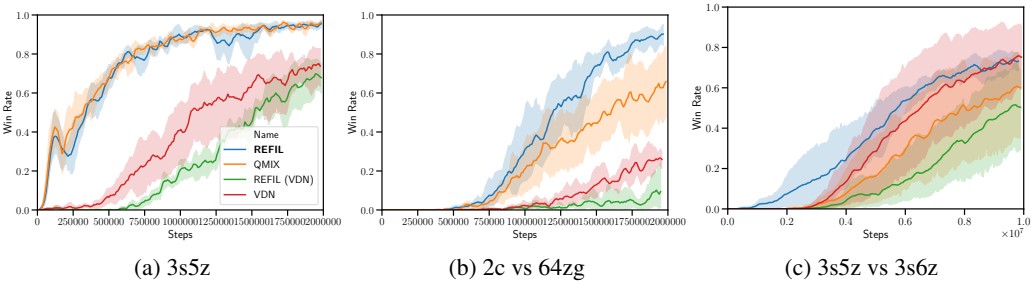

(a) 3s5z  (b) 2c vs 64zg  (c) 3s5z vs 3s6z

Figure 7: Results on SMAC.

Results can be found in Figure 7. While we expect **REFIL** to be most effective in the setting of varying types and quantities of agents, we still find that it improves on QMIX in 2 of the 3 scenarios tested. In the standard SMAC benchmarks, we find our approach is able to match or outperform the best baseline across all settings. Specifically, our factorization method (which builds on QMIX) improves the performance of QMIX in 2 of 3 settings tested. As far as we are aware **REFIL** outperforms all reported results in the literature on "2c vs 64 zg" (clasified as a "hard" task in SMAC). The relative improvement over QMIX, combined with the fact that it does not ever appear to hurt performance, indicates that the benefits of our method are not limited to settings with varying types and quantities of agents, though the positive effects are more pronounced there.

