# OpenReview forum: "Randomized Entity-wise Factorization for Multi-Agent Reinforcement Learning"
_ICLR.cc/2021/Conference — Reject_

### Official Review · AnonReviewer4 · 2020-10-28
**A more serious discussion about [Agarwal et al., 2020] is expected.**

**Rating:** 5
**Confidence:** 4

**Review:**

This paper introduces a randomized entity-based attentional mechanism to regularize the observation space for efficient multi-agent reinforcement learning. Specifically, the authors expect that their method can help agents focus on entities that are relevant to their decision-making process. The aim of the paper is well-positioned in multi-agent settings and are expected to help improve performance by exploiting the loosely coupled structure of multi-agent tasks (although the authors do not explicitly model the decision dependency among agents). However, I have some doubts about whether the proposed method can address the target of the paper.

Intuitively, in multi-agent settings, the number of entities is at least the number of agents, say O(n). O(n) is a quite optimistic estimation because we even do not consider other entities. If the authors want to find an optimal bi-partition over the entity space for each agent, the search space is at least O(2^n), which grows exponentially with the number of agents. To design an efficient search algorithm over such a large space needs to take advantage of some well-designed inductive bias or heuristics. The authors use a random strategy here, which, in my opinion, is not sufficient to guarantee a satisfactory solution. There is no denying that randomization can give a good solution in some cases, but this can not be held as a general rule. Even if the bi-partition structure is trained end-to-end, I also suspect that learning such a structure is not easier than learning from scratch.

Additionally, I think the authors largely ignore the contribution of a related work [Agarwal et al., 2020]. Although they cite this paper in Sec. 5, unlike what is stated in this paper, the main contribution of [Agarwal et al., 2020] is a GNN-based attentional mechanism over entity spaces. They propose to let agents learn to attend to different entities under different observations. In this way, the target of [Agarwal et al., 2020] and this paper largely overlap. I was expecting that the authors provide a thorough comparison with [Agarwal et al., 2020] in their experiments. If the authors can demonstrate that their method can outperform [Agarwal et al., 2020], I will consider improve my rating.


[Agarwal et al., 2020] Agarwal, A., Kumar, S., Sycara, K. and Lewis, M., 2020, May. Learning Transferable Cooperative Behavior in Multi-Agent Teams. In Proceedings of the 19th International Conference on Autonomous Agents and MultiAgent Systems (pp. 1741-1743).

---

> ### Author Response · Authors · 2020-11-18
> **Thank you**
>
> Thank you for the helpful comments. We believe there are some slight misconceptions contributing to the reviewer’s doubts that we would like to clarify. Specifically, the statement “the authors want to find an optimal bi-partition over the entity space for each agent” is not accurate. We do not mention the notion of optimal groups nor the notion of search in our paper, and this is intentional. In fact, even when we provide the exact group of “optimal” entities to each agent as an attention mask (REFIL (Fixed Oracle) in Figure 3b), we find that performance does not improve in our group matching domain. The randomization is not a limitation of our approach, but in fact a feature. Our method samples randomly, not as a search method, but rather to learn to predict value functions by assembling pieces (sub-groups) into a whole. In doing so, our model will identify timesteps where agents’ utilities are purely captured from within group utilities by assigning low value to $Q_O$ (utility of out-of-group interactions). As such, the model can then use the utilities learned from this sub-group to inform its prediction the next time it is encountered. Of course it is possible to train a parametrized model that can reduce the set of sub-groups we consider; however, from the results of REFIL (Randomized Oracle) in Figure 3b, we do not currently possess any evidence that this would be helpful for the scale of environments we consider. It is possible that such an approach would provide benefits in a larger scale setting (e.g., 100 agents).
>
> We apologize for the relative lack of discussion/misrepresentation regarding Agarwal et al 2020. We believe we have ameliorated this issue in the updated draft and are happy to receive any more feedback to improve the discussion. With respect to the relation of our work to Agarwal et al, we note that, as the reviewer states, the main contribution of Agarwal et al is “a GNN-based attentional mechanism over entity spaces”. While we use multi-head attention as the method to aggregate information across entity spaces, we do not position this as a contribution of our work. Instead we focus on developing an auxiliary training scheme to improve generalization in settings with diverse entity configurations. The main difference between the Entity Message Passing (EMP) proposed by Agarwal et al and the attention mechanism we use is that EMP allows for several steps of message passing, while we only use 1.
>
> We have provided a comparison to Agarwal et al. in our updated experiments. We incorporated the EMP structure into QMIX in order to compare to the Multi-Head Attention module that we use. We use QMIX as the base algorithm, rather than the PPO algorithm used in the paper for the following reasons: 1) As the reviewer states, the main contribution of this paper is “a GNN-based attentional mechanism over entity spaces.” By using QMIX, we can directly compare this method for entity aggregation to the method we use without any confounding factors (i.e.. the base algorithm). 2) PPO has not been well validated in the MARL literature to perform well on SMAC tasks. In general, actor-critic methods have not fared well to date in this domain.
>
> We find that EMP does not improve performance over QMIX (Attention) in the settings we test. We believe this occurs for the following reason: EMP is particularly useful in cases where partial observability is a significant hindrance to successful completion of the task. While SMAC contains partial observability, it is typically not an issue for learning (agents almost always see the entities that are relevant to them). In this way, the target of EMP and REFIL are almost opposite. EMP seeks to aggregate as much information as possible, while REFIL aims to discard information when possible (in training only) in order to improve generalization. We hope that these additional comparisons satisfy the reviewer’s concerns, and we are happy to answer any follow up questions.

---

### Official Review · AnonReviewer3 · 2020-10-28

**Rating:** 5
**Confidence:** 3

**Review:**

This paper proposes an observation factorization method to avoid the influence of the irrelevant part on value estimation. Specifically, they design an entity-wise attention network with a masking procedure. This network is used to filter the irrelevant part of the original observation of each agent. Then the output is used to estimate the individual q-value, as well as input to the mixing network to generate the Q_tot. Two kinds of Q_tot are trained together by combing two loss functions linearly with a hyper-parameter. Experimental results show REFIL combined with QMIX surpasses vanilla QMIX and VDN in several SMAC scenarios.

This paper is related to the topics of ICLR. However, I think the related work is not sufficient to cover the background. More detail comments can be found below.

*****Some specific comments:*****

It is not clear that what is the initialization of two masks, and how to update the masks.

The authors mentioned there are two groups of entities. However, the entity type is also unclear. I guess SMAC only contains two entity types: alive agents and died agents? How to represent an entity inactive?

One question is why just consider two kinds of groups, what would happen if there exist more than two groups for all entities. In SMAC or soccer, it does contain more than two common patterns. Furthermore, it seems that the masking procedure is hard to extend to the situation with a larger number of groups.

Actually, I think REFIL is similar to ROMA [1] and ASN [2] in different ways. First, REFIL considers two kinds of groups corresponding to a simple version of ROMA which has two roles. Second, REFIL does the same thing as ASN that learns the value estimation by considering a more useful part of the observation. ASN directly divide the observation based on the action semantics, while REFIL tries to learn a suitable observation factorization through entity-wise attention with masking. However, these two very relevant works are not discussed and compared in this paper.

Some suggestions,

I think current experiments could not well support motivation. If authors show some examples in SMAC that what kinds of common patterns agents learn would be better to support this idea.

Since REFIL can be integrated into current MARL algorithms, it is better to consider more recent published MARL methods as baselines, such as QTRAN, QATTEN, QPLEX.

[1] Roma: Multi-agent reinforcement learning with emergent roles. ICML. 2020.

[2] Action Semantics Network: Considering the Effects of Actions in Multiagent Systems. ICLR. 2020.

---

> ### Author Response · Authors · 2020-11-18
> **Thank you**
>
> We thank the reviewer for their useful feedback. We have significantly extended our experiments section in order to provide comparisons to several suggested more recent methods (QTRAN, QATTEN, ROMA), and we find that these do not improve performance in our dynamic entity domains, highlighting the unique challenge these domains pose. We have also added a diagram (Figure 6) demonstrating an example of common patterns that our method leverages across different episodes where the overall set of entity types is different. We will now respond to the reviewer’s specific concerns.
>
> “It is not clear...the initialization of two masks, and how to update the masks.”
> We apologize for the confusion. The masks are initialized such that agents can only see the entities within their group. The groups themselves are sampled as random splits of the entities (Bernoulli(p) for each entity, where p is sampled once per episode in training from Unif(0,1)). We split into two groups in order to induce a sub-group distribution that is uniform over all possible sub-groups (splitting into more groups would create a bias towards smaller sub-groups).
>
> These masks are not updated but rather held fixed for the duration of an episode sampled for training. We emphasize that the goal of the work is not to learn how to split entities into independent groups. In order to do that we would need some sort of parametrized model that predicts the relevant entities for each agent. Instead, our goal is to split entities into random groups such that we learn reusable components that can be transferred better across settings where the types and quantities of entities are variable.
>
> “The authors mentioned there are two groups of entities... entity type is also unclear...How to represent an entity inactive?”
> It appears that two separate concepts are being mixed here. The two groups in our method do not have a fixed semantic meaning. As mentioned previously, these are randomly sampled such that we learn reusable components for improved value function transfer across diverse settings. The set of entity types, on the other hand, is fixed and provided by the environment. In SMAC the entity types correspond to unit types (Stalker, Zealot, etc.). In general, entity types define the dynamics/capabilities of an entity. Entities are represented as inactive by masking their observability (as is done in the standard SMAC).
>
> “why just consider two kinds of groups...it seems that the masking procedure is hard to extend to the situation with a larger number of groups.”
> As previously mentioned, the groups do not have fixed semantic meanings. Our method identifies independence between groups by assigning low value to the $Q_O$ (out of group interaction) terms when independence exists in the randomly sampled groups. By sampling randomly, our model is able to discover any possible number of sub-groups, not just two, and can even discover independent sub-groups that exist temporarily (since the value assigned to $Q_O$ is dynamic at each timestep). Finally, the masking procedure can be extended to consider a larger number of groups trivially, without any additional computation. This is described in the last paragraph of section 3.

---

> > ### Author Response · Authors · 2020-11-18
> > **(continued)**
> >
> > “REFIL is similar to ROMA [1] and ASN [2] in different ways...”
> > We did not include comparisons/discussions of these works initially, as we did not view them as closely related; however, we have added them to the related work section. We would like to clarify the differences between these works and our approach.
> >
> > ROMA: We reiterate that our method does not only consider “two kinds of groups” but rather randomly splits agents into two groups such that we learn estimated returns in sub-groups that can potentially be found in other scenarios. We do not learn any representation space for these groups. ROMA, on the other hand, specifically learns a role embedding space such that individual agents adapt their behavior based on their situation. In contrast, our groups are not learned via a parameterized model and contain an arbitrary number of agents/entities. The goal is not to learn distinct roles, but rather to discover existing patterns and share information across settings when these patterns are replicated. We find that ROMA struggles to learn in our environments. The challenge of learning roles is most likely exacerbated in the setting where unit types are different at each episode.
> >
> > ASN: We believe that it is a mischaracterization to state that REFIL does “the same thing as ASN.” ASN utilizes domain knowledge to directly map segments of the observation space to the corresponding action. This is achieved through a modification of the utility network architecture. Our method, on the other hand, assumes no domain knowledge, and does not modify the observation space during execution. The masking of observable entities only occurs during the computation of the auxiliary loss during training. We do not view ASN and REFIL as competing approaches, and in fact, they could potentially be combined, just as ASN is combined with several different approaches in their paper.
> >
> > We hope that these comments as well as our follow up experiments have addressed the reviewer’s concerns about the paper. We are happy to answer any follow up questions.

---

### Official Review · AnonReviewer2 · 2020-10-29
**Novel approach to factorize entities in a multi-agent env with supportive empirical evidence**

**Rating:** 7
**Confidence:** 4

**Review:**

The paper proposes a method for randomized factorization of multi-agents for efficient learning. The idea is inspired by the presence of irrelevant entities present in an agent's observational view and how removing those could aid the learning process.
Agents are randomly divided into groups so that an agent can separately measure the influence of entities present in the same group and the entities present in the other groups. Since the groups are randomized, this helps the agent to create groups of variable size based on its utility prediction of in-group and out-of-group entities.

Although the paper only uses two groups for derivation and experiments, it claims that the same method can be applied to more than two groups but is yet to be demonstrated.


The paper is easy to read and the figures are excellent and self-explanatory. The environments chosen are also indicative of the importance of each component. The authors also found that randomized factoring performs better empirically than using domain knowledge while the state-of-the-art lies in the combination of the two. Further, the idea of training variable-sized hypernetwork is quite fascinating and fits well in the overall framework.

Some questions:
Figure 3b does not show results till convergence, please put the entire plots.

I think it would be better if the SMAC setup is explained in more detail. I couldn't understand how the tagging is done (in Appendix). Does it deterministically tie an action to an enemy?

QTRAN has been shown to perform better than QMIX in competitive domains. I would encourage the authors to compare the results with this baseline too.

---

> ### Author Response · Authors · 2020-11-18
> **Thank you**
>
> We thank the reviewer for their feedback and comments. We have included the results for Figure 3b while running for an additional 1M timesteps. We noted that the separation between REFIL and REFIL (Randomized Oracle) grew smaller in these new runs, so we ran an additional 12 runs, finding that these two methods perform similarly on aggregate. This is interesting, as it implies that the use of domain knowledge in selecting sub-groups is not particularly useful, and the main benefit of our method comes from randomly selecting sub-groups. Please see our response to reviewer 1 for more detail regarding why this might be the case.
>
> We apologize for the confusion in the description of the SMAC setup and have added additional detail regarding the tagging procedure. In short, each enemy is randomly assigned a tag at the start of each episode, and each agent has an action corresponding to attacking any possible tag. Agents observe the tags of the enemies. In the standard SMAC setting, there is a separate action for attacking each distinct enemy, but we cannot use this setup since the set of enemies changes at each episode.
>
> We have included experiments on QTRAN as well as several other more recent follow ups to QMIX, finding in general that they do not improve performance in our setting, highlighting the unique challenge of learning in settings with variable groups of entities.

---

### Official Review · AnonReviewer1 · 2020-10-30
**Interesting problem, simple but effective method**

**Rating:** 6
**Confidence:** 4

**Review:**

Summary: This paper proposes to incorporate a masked attention mechanism in QMIX for value function factorization to disentangle value predictions from irrelevant agents/entities. The masking is based on a random sampling from the whole set of agents to from random subsets, based on which it can compute within-group and without-group Q-functions. The method is able to handle varying types and number of agents. The paper conducts experiments on a simple game to understand the effect, and then test on 3 SMAC games, which shows the effectiveness of the proposed REFIL method.

Strong points:
- The paper is well-written and clear.
- The paper studies an important topic in MARL, i.e., how to deal with varying types and number of agents, and propose a simple yet effective approach, which incorporates attention mechanism in QMIX with random masking.
- Experiments on several games illustrate the effectiveness of the method, with proper ablation study to understand the importance of each component (attention, mixing network, random masking).

Concerns:
The main focus of the paper is to “disentangle value predictions from irrelevant entities”. However, not only does REFIL relies on Q_I (within-group Q-function), but also it relies on Q_O (without group Q-function). If the agent successfully learn this neglect of irrelevant entities, focusing on Q_I would be enough, without triggering additional computation of unnecessary Q_O. Could authors better explain this? In addition, consider the breakaway example, as the attacker has only to focus on the goal keeper, is the random sampling scheme from all agents effective compared with counterparts that only need to focus on the goalkeeper? Could authors conduct additional experiments on football to better support the claim?

---

> ### Author Response · Authors · 2020-11-18
> **Thank you**
>
> We thank the reviewer for their valuable comments and for providing an opportunity to clarify our approach, specifically with respect to the in-group ($Q_I$) and out-group ($Q_O$) Q-functions. The key reason that necessitates the use of $Q_O$ is the fact that we are *randomly* sampling entity groups. As such, we do not know a priori whether the entities outside of a particular agent’s group can be ignored and must always potentially consider $Q_O$; however, in the case that entities outside of an agent’s group *can* be ignored, the mixing network assigns those $Q_O$ factors to have near-zero value (we find this empirically to be the case in our group matching game when we feed in the ground-truth groups as masks to our method trained with random groups). In this case, our model is learning to ignore irrelevant entities. As such, a potential conclusion to make is that our method relies on frequently sampling “correct” splits of entities into separate independent groups in order to learn effectively; however, this conclusion is inaccurate.
>
> This statement leads into the reviewer’s second question regarding whether random sampling is effective. In fact, this question is exactly what motivates our experiments in the group matching game. Here we find that simply providing the ground truth regarding which entities are relevant to each agent as a mask (REFIL - Fixed Oracle) does not improve performance over QMIX. As such, random sampling is in fact *crucial* for the improved performance we see in REFIL. We provide the following analogy to understand why this might be the case: consider the task of image captioning and assume we have access to the bounding boxes for all entities in the image (though we may have no semantic understanding of what these entities are). The full image is analogous to the full set of all entities’ states in our setting. Let’s say we have an image with a dog sitting in a room playing with a ball and a cat sitting on a windowsill in the background. This image contains the entities: dog, ball, cat, and windowsill. It can be effectively captioned by considering the following entity groups independently: (dog, ball), (cat, windowsill). Now let’s say we randomly sample the groups (dog), (ball, cat, windowsill) while training the captioning model. In this case, we have missed out on interactions between the ball and dog which will need to be accounted for by the interaction term; however, we can still learn about the concept of what a dog is. This concept may come in handy later when we see a dog in another image that is otherwise completely novel. Our method works in an analogous manner, where we learn to decompose value functions into pieces such that these pieces can be reassembled to estimate values in settings with different groups of entities. The main benefit of our method is that it allows our models to explore *all* possible independence relationships between entities, without specifically enforcing that the groups we sample *must* be independent.
>
> We have also added a diagram to our experiments of a common pattern that emerges in StarCraft (Figure 6) as well as a discussion of how our model can leverage such shared patterns across diverse settings.
>
> We hope that this discussion helps clarify the reviewer’s concerns, and we are happy to answer any follow up questions.

---

### Author Response · Authors · 2020-11-24
**Summary of Updates**

We would like to thank all reviewers for their useful and insightful feedback. We have made extensive additions to the experimental section of the paper based on reviewer suggestions, and we will summarize these additions here:

* Additional comparisons
    * Added Table 1 which provides a comparison of the attributes of all methods tested.
    * We have adapted several MARL methods suggested by the reviewers, using the original authors' open source code, to our variable entity setting by using the attention models we use in our own approach. These include QTRAN [1], Qatten [2], and ROMA [3]. We find that these methods in general are not competitive with our approach. These results provide additional evidence that the challenges presented by variable entity domains are unique to those in standard fixed entity MARL tasks. Specifically, our work improves the ability for agents to transfer their experiences across different entity configurations.
    * As suggested by reviewer 4, we have included experiments comparing to an alternative mechanism [4] for aggregating information from variable sets of entities. We find that this method does not significantly improve performance over the multi-head attention mechanism we use.
* We have extended the discussion of related work as suggested by reviewers.
* We have added Figure 6, which provides a visual example of common patterns of behavior that emerge across variable entity configurations in StarCraft. This figure should help readers gain a more intuitive understanding of why our imagined sub-group sampling method improves performance over the baselines. Specifically, our method enables the direct reuse of utility predictions within sub-groups of entities ($Q_I$) across various global configurations where similar sub-patterns emerge.

We have also addressed each specific question from the reviewers in our individual replies. If any questions or concerns remain, we are happy to address them before the end of the discussion period today.

[1] Son, Kyunghwan, et al. "QTRAN: Learning to Factorize with Transformation for Cooperative Multi-Agent Reinforcement Learning." International Conference on Machine Learning. 2019.

[2] Yang, Yaodong, et al. "Qatten: A General Framework for Cooperative Multiagent Reinforcement Learning." arXiv preprint arXiv:2002.03939 (2020).

[3] Wang, Tonghan, et al. "Roma: Multi-agent reinforcement learning with emergent roles." Proceedings of the 37th International Conference on Machine Learning. 2020.

[4] Agarwal, Akshat, et al. "Learning Transferable Cooperative Behavior in Multi-Agent Teams." Proceedings of the 19th International Conference on Autonomous Agents and MultiAgent Systems. 2020.

---

### Decision · Program_Chairs · 2021-01-07
**Final Decision**

**Decision:**

Reject

**Comment:**

This paper proposes an attention based technique to focus on relevant entities in multi-agent reinforcement learning.  While the effectiveness of the proposed method is demonstrated on some tasks, there remain major concerns including the following:
1. It is not sufficiently convincing that the proposed method performs well in more complex domains
2. Novelty over Agarwal et al. and MAAC is rather minor